# Review of Canine Lymphoma Treated with Chemotherapy—Outcomes and Prognostic Factors

**DOI:** 10.3390/vetsci10050342

**Published:** 2023-05-11

**Authors:** Peter Bennett, Peter Williamson, Rosanne Taylor

**Affiliations:** 1The Sydney School of Veterinary Science, Faculty of Science, The University of Sydney, Camperdown, NSW 2006, Australia; 2The University of Sydney Nano Institute, The University of Sydney, Camperdown, NSW 2006, Australia

**Keywords:** dog, cancer, lymphoma, chemotherapy, prognostic factors, remission, survival

## Abstract

**Simple Summary:**

Canine lymphoma is a common cancer of one of the white blood cells that is commonly treated with chemotherapy. Many papers have been published about this disease and treatment, but few comprehensive reviews of the outcomes reported after chemotherapy, and the prognostic factors that influence the outcomes, have been published. Knowledge of the potential outcomes for this disease is important for veterinarians as they guide dog owners in the decision making about treatment options. This review evaluated the papers published on this topic and found that there is a wide variation in outcomes reported, as well as the factors that influence the outlook. Additionally, while individual papers have shown a much better survival, over the last 40 years, the overall improvement in outcome has been modest at best. The use of uniform reporting in the studies is needed, and novel approaches are likely needed to significantly improve the outlook for dogs with this disease.

**Abstract:**

While canine lymphoma is a relatively common and important disease seen by veterinarians, there are limited comprehensive reviews of the literature regarding the remission and survival times following chemotherapy, and the associated prognostic factors. This comprehensive thematic review covers the available veterinary literature covering treatment outcomes and identified prognostic factors. A lack of standardised approaches to evaluate and report the outcomes was identified, including factors that would alter the duration of responses by weeks, or occasionally months. After publication of the suggested reporting criteria, this has improved but is still not uniformly applied. The prognostic factors included for evaluation varied from as few as three to seventeen, with over 50 studies using only univariate analysis. Individual papers reported much longer outcomes than others, but assessing the outcomes overall, there has been minimal change over the last 40 years. This supports the belief that novel approaches for lymphoma therapy will be required to substantively improve outcomes.

## 1. Introduction

Canine lymphoma is common in clinical practice with an estimated frequency of about 20/100,000 dogs [1]; however, there has been limited critical review of the outcomes and prognostic factors. The disorder is most often seen in dogs as a systemic disease with nodal, liver, and splenic involvement that can include non-lymphoid sites and the bone marrow. In people, a range of subtypes of lymphoma have been reported, which can affect the expected outcomes and preferred treatment options [2]. Many of these subtypes have also been identified in dogs but with limited information on the impact on outcome and treatment options [3,4]. The mainstay of treatment for canine patients is chemotherapy, with improved outcomes reported when combination protocols are used. The most frequently used protocols in the veterinary setting are based on the CHOP (cyclophosphamide, doxorubicin (hydroxydaunorubicin), vincristine (Oncovin), and prednisone/prednisolone) protocol [5]. These were adopted from human medicine with modifications based on differing drug pharmacology and acceptance of the risk of adverse events. A veterinarian in practice needs access to the evidence for likely outcomes following treatment, including the influence of prognostic variables, to better discuss the options with clients. 

While over 1000 papers have been published on canine lymphoma, there is variation, however, in the patient and lymphoma details provided, as well as the inclusion/exclusion criteria used, and inconsistency in the inclusion of potential prognostic factors. Many summaries have been published in textbooks and review articles, but few provide a critical review of the studies [6], instead reporting study outcomes or prognostic criteria, without consideration of the variations in methodology and reporting criteria within the studies. Until 2010 there was not a recommended consensus on the definitions to be used for reporting responses, outcomes, remission, and survival durations, leading to variations in how outcomes were reported. Since the publication of response evaluation criteria for nodal lymphoma from the Veterinary Cooperative Oncology Group (VCOG) in 2010 [7], there has been increased uniformity, but not across all studies. 

Full classification of the lymphoma types in veterinary studies is often lacking, including in many recent studies that do not classify the disease beyond immunophenotype. This contrasts with studies of lymphoma in people where histological and genomic classifications are widely used as inclusion criteria in treatment studies [8]. People with lymphoma are routinely treated with immunotherapy with anti-CD20 monoclonal antibodies with high-risk B-cell lymphoma [8]; being able to clearly identify similar high-risk groups of dogs is required to justify the use of similar options. As veterinarians adopt some of these methods, better information will be important to enable the selection of suitable cases, as well as identifying the subtypes, both histological and molecular, that stand to benefit most from more, or less, aggressive interventions. 

The authors’ aim in this review was to evaluate the veterinary literature and report on the variability in the current information that is available, to identify the gaps that are revealed, and to highlight the need for best practice in the reporting of lymphoma studies in the veterinary literature. This information will highlight the deficiencies in how canine lymphoma treatment studies have been reported.

## 2. Materials and Methods

A search was performed in the Web of Sciences (all databases) using the terms (dog or canine) and (lymphoma or lymphosarcoma) for all years until the end of 2021. All papers were accessed where available either electronically or hardcopy. Most papers in languages other than English were not available for review. The papers were reviewed for outcomes after the use of chemotherapy, if additional therapies were used, the inclusion/exclusion criteria for the study, and the details of the statistical analysis. The data extracted included if the case recruitment was prospective or retrospective; if cases were naïve (no prior chemotherapy), relapsed, a mixture, or not specifically stated; if the immunophenotype of the cases was known; if histological classification was included; if specific anatomic locations were included/excluded; the chemotherapy protocol(s) used; and if cases with early relapse/death were excluded. When the preceding information was not available, this was recorded. 

Chemotherapy protocols were divided into CHOP-like, COP-like (cyclophosphamide, vincristine, and prednisone/prednisolone), MOPP-like (Mechlorethamine, vincristine, procarbazine, and prednisone), other combinations, and single agents. The CHOP-like protocol included cyclophosphamide or chlorambucil, a vinca alkaloid, an antitumour antibiotic (doxorubicin, mitoxantrone, actinomycin-D, epirubicin), and usually prednisone/prednisolone, with or without L-asparaginase. COP-like protocols included cyclophosphamide or chlorambucil, a vinca alkaloid, and prednisone/prednisolone with or without L-asparaginase. MOPP-like protocols included mustargens/CCNU/bleomycin, vincristine/vinblastine, procarbazine, and usually prednisone/prednisolone. Other combinations were combinations of two or more drugs that did not fit the above criteria. There were many single agents used, most often doxorubicin. The use of prednisolone/prednisone was variable across these protocols and was not always clearly indicated if used as a standard dose or duration, leading to its use not altering the group allocation. Some studies included a variety of treatments and reported only the overall outcome, which were included in a treatment group named as mixed. 

Response to treatment definitions were extracted when defined in the studies. The rates of complete remission (CR), partial remission (PR), progression free or remission times, survival times, and one-, two-, and three-year survival percentages were recorded where available. When both a CR and PR rate, or the CR rate was 100%, an overall response rate (ORR) was calculated if not reported. When outcomes for dogs achieving a CR or PR were reported separately, this was documented. In some papers that compared different protocols, the outcome information was identified for each protocol, in addition to the overall. When the protocols were similar, varying only in duration, or had drugs substitutions but remained within the definitions previously given, they were included as a single protocol if there was no significant difference in outcome. When there was a significant outcome difference between the treatment protocols, they were individually included. 

Specific anatomical localisations of lymphoma were separated from multicentric or nodal lymphoma. In studies without designation of anatomical location, the cases were designated as mixed, with the majority likely to be multicentric or nodal. Studies reporting relapsed or resistant lymphoma were considered separately to naïve cases. 

Prognostic factors included in the study were evaluated. Each paper was reviewed to identify which factors were included in the analysis, and the details recorded included if univariate or multivariate analysis was performed. Substage was recorded as indicated by the attending veterinarian when included in the record or based on the history and examination findings. Some variables reported in this review were combined. Prednisolone and prednisone were considered equivalent if prior use or inclusion in the protocol was evaluated in the study. Treatment as a prognostic variable referred to differing protocols, either drugs used or duration/dosage of the drugs. Coagulation included clotting parameters or thromboelastography changes. For neutrophils and monocytes, both cytophilia and ratios were combined. The genetic/molecular markers group includes genotype analysis, specific molecular markers, and karyotype evaluation. If the Section 2 did not clearly list all the factors that were included in the statistical analysis, the paper was scanned to identify as many as possible from the results and discussion. 

The statistical methods used in the papers varied. When assessing prognostic factors, only studies assessed with multivariant analysis were included in this paper. The criteria for a factor to be included in the multivariant analysis was recorded. When only univariate analysis was included in the paper, the information was tabulated in the Appendix A in this paper.

All results were tabulated in Excel spreadsheets and statistical analysis was limited to descriptive statistics (median, mean, range). To simplify the discussion in this paper, remission was used to designate all variations of remission and progression free survival, while survival was used to designate the various definitions of overall survival including lymphoma specific survival. Correlation evaluations were performed in IBM SPSS version 26 using the Pearson’s coefficient. 

Each journal article was designated as a paper; within each paper, each individually assessable population of cases was designated as a group. The dogs included in each group were designated as cases.

## 3. Results

The initial database search found 5882 papers including duplicates. These were reviewed for those describing clinical disease in dogs, refining this to 1179 individual papers. Evaluation of these papers identified 156 from which treated lymphoma outcome information was available, involving 211 groups and a total of 10,567 cases. There were 112 papers identified in which prognostic variables for outcome were reported. 

Treatment with vaccine or other immunotherapy was reported in seven papers with outcomes for a control group treated only with chemotherapy, with or without a placebo, reported with the non-immunotherapy groups included [9,10,11,12,13,14,15]. Eleven papers included 13 groups that were treated with radiation therapy, hyperthermia, or bone marrow or stem cell transplantation, with these groups excluded from the review [16,17,18,19,20,21,22,23,24,25,26]. Two papers were excluded as they reported on the outcome after surgery alone for splenic lymphoma [27,28]. 

### 3.1. Treatment Response, Remission Duration, and Survival Definitions

Treatment response was defined in the Section 2 of most papers but not all, and in the majority, it was based on physical examination with variably implemented imaging. There were differences identified with most papers defining complete remission (CR) as complete absence of visible disease, though assessed as a >75% reduction in two studies [29,30] and >50% in another [31]. Partial remission (PR) was most often defined as a >50% decrease in tumour volume, though how this was determined was not often defined, stable disease (SD) as between a <50% decrease and a <20–30% increase in tumour volume, and progressive disease (PD) defined as new lesions or a greater than 20–30% increase in tumour volume. After 2010 there was an increase in the use of modified RECIST (response evaluation criteria in solid tumours) measures where target lesions were identified, and the sum of the longest diameter was used as the measure. The detail of the system most often used is described in the VCOG consensus statement [7], though some studies used very similar definitions but did not reference the VCOG paper. A summary of the treatment response definitions is included in Table 1. 

The timing of the assessment of the dogs and the minimum duration of response required to be classified as such varied between papers. For many papers, neither the timing nor the duration of response were defined, and the strongest response was recorded and used, or the response at a specific day in the protocol was used (day 22) [32], after one cycle of treatment [33]. The response had to be sustained for between 7 and 28 days (seven days in one paper [34], 14 days in three papers [35,36,37], 21 days in 19 papers [5,33,38,39,40,41,42,43,44,45,46,47,48,49,50,51,52,53,54], and 28 days in four papers [12,55,56,57]). There were very few papers where uniform staging was performed to confirm the response indicated. 

There was inconsistency in the literature for the starting point of remission duration. This varied from the date of diagnosis, the date treatment commenced, the date when the greatest response was achieved, the end of induction, the end of the first cycle, the end of the chemotherapy protocol, or it was not clearly defined. In addition, standardised terminology was not used to describe the duration of the first remission with multiple terms used for similar measures. A list of the terms and the number of studies utilising them is in Table 2. Whether censoring of animals that were alive or lost to follow-up was used was not consistently documented in the papers. 

Relapse and progression were not defined consistently. Many papers did not offer a definition, others used the identification of progressive disease as meaning relapse. A small number of papers required a return to the nodal size at the time of diagnosis and/or new nodal enlargement for a patient to be considered to have relapsed.

Survival time definitions were variable with many not specifying the start date. When defined, it varied between the date of diagnosis, the date of starting chemotherapy, and the date of strongest remission. The majority included all causes of death, while lymphoma-specific survival was used in a smaller number. Most papers reported both a remission and survival time (88 papers); 43 reported only a remission time and 25 only survival times.

### 3.2. Outcomes for Chemotherapy Treatment

The outcome for chemotherapy treatment was reported in 173 papers including 197 groups and 10,301 cases. Of these, 136 papers with 151 groups and 9191 cases reported a combination protocol with the remainder having a single agent protocol. There were two groups from two papers treated only with prednisone or prednisolone [58,59]. Of the papers reporting on chemotherapy outcomes, 65 including 81 groups had prospective recruitment of patients, 91 with 110 groups were retrospective, and in four papers with six groups [36,59,60,61], it was not clearly identified. Four papers included both prospective and retrospective groups [22,62,63,64]. Outcomes for cutaneous lymphoma, both epitheliotropic and non-epitheliotropic, were reported in six papers with nine groups [34,42,65,66,67,68]. There were four papers with four groups with gastrointestinal tract lymphoma treated with chemotherapy [41,54,69,70], and one group from a paper reported on nasal lymphoma treated with chemotherapy [18]. 

There was variability in the exclusion criteria regarding treatment prior to the use of chemotherapy. Only patients that had not received prior chemotherapy were included in 85 papers with 107 groups involving 5269 cases [5,9,10,12,15,18,30,36,38,43,49,54,55,56,57,60,62,63,69,71,72,73,74,75,76,77,78,79,80,81,82,83,84,85,86,87,88,89,90,91,92,93,94,95,96,97,98,99,100,101,102,103,104,105,106,107,108,109,110,111,112,113,114,115,116,117,118,119,120,121,122,123,124,125,126,127,128,129,130,131,132,133,134,135,136]; 25 papers with 27 groups and 1116 cases included only relapsed lymphoma [33,37,39,44,45,46,47,48,50,53,115,137,138,139,140,141,142,143,144,145,146,147,148,149,150], eight papers with 292 cases in 10 groups had a mix of naïve and relapsed cases [67,122,151,152,153,154,155,156], and the remaining 39 papers with 53 groups and 3461 cases [11,13,14,22,29,31,32,34,35,40,42,51,52,58,59,61,65,66,68,70,157,158,159,160,161,162,163,164,165,166,167,168,169,170,171,172,173] did not specifically state if prior treatment was excluded, suggesting that the cases included were naïve or predominately so. 

Specification of the anatomical location as an exclusion criterion varied across the literature. There were papers that specifically excluded cases with bone marrow [30,102], cutaneous [116,118,126,168], gastrointestinal [40,126], or central nervous system [168] involvement, while in others, these were not specifically stated as exclusion criteria. Stage V disease was excluded in two papers [102,132], stage V only was included in three [113,117,170], with stages I and II excluded in 11 papers [13,15,30,63,77,102,114,126,152,163]. Cases with hypercalcaemia were excluded in two papers [111,167]. In some papers, only patients achieving a remission, sometimes restricted to complete remission, were included in the outcome analysis [9,22,43,81,114].

### 3.3. Remission Times, Survival Times, and Response Rates for Multicentric/Nodal Lymphoma 

The outcome for multicentric or generalised nodal lymphoma treated with a CHOP-like protocol was reported in 43 papers with 54 groups and 4382 cases [5,12,13,15,22,38,40,43,55,56,58,64,76,83,85,88,90,92,95,97,98,99,100,102,107,110,111,112,113,118,120,123,124,126,132,134,136,166,168,169,172,174,175]. Nine papers included only dogs with B-cell lymphoma [12,13,15,55,56,76,88,102,124], three included only T-cell lymphoma [100,123,126], and the remainder had a mix of both immunophenotypes. The group sizes ranged from 7 to 775 cases. The cases were identified as naïve in 33 papers including 39 groups and 2493 cases, with the remainder not clearly stating this in the methods. Remission only was reported in six papers [38,43,64,123,136,172], survival only in five papers [58,85,97,98,111], and both in the remainder. The reported outcomes are listed in Table 3. 

There were an additional 26 papers including 33 groups and 1606 cases [11,32,35,36,52,57,62,78,82,84,86,87,95,96,101,104,105,106,119,128,129,133,136,151,161,171] treated with a CHOP-like protocol where the location or distribution of the lymphoma were not clearly defined. Two papers included only B-cell lymphoma [84,161] with the remainder either not specifying the immunotype or including a mixture. The group sizes ranged from 8 to 134 cases. It is likely that most of the cases in these papers were multicentric or nodal. One paper reported only survival time [78], four only remission duration [32,86,133,136], and one reported only remission rates [82], with the remainder reporting both remission and survival durations. The outcomes are listed in Table 3. 

There were 21 papers with 27 groups that included patients treated with a COP-like protocol [5,9,10,14,29,59,60,75,80,81,93,109,114,148,157,158,163,164,165,167]. The distribution of the lymphoma was stated as multicentric in about half the papers, with the remainder not clearly stating. The group sizes ranged from 7 to 147 cases. One study included only dogs with B-cell lymphoma [10], while the remainder had a mix of both immunophenotypes. Remission only was reported for four papers [86,148,158,165], survival only in one paper [93], and both for the remainder. The outcomes are listed in Table 3. 

Eleven studies included 14 groups and 481 cases with multicentric or nodal B-cell lymphoma treated with a CHOP-like protocol [12,13,15,55,56,76,84,88,102,124,161]. The group sizes ranged from seven to 104. All the studies reported remission and survival times. The outcomes are listed in Table 3. 

T-cell multicentric or nodal lymphoma treated with chemotherapy was reported in 8 papers with 9 groups and 476 cases. Of these, 3 papers with 3 groups included 107 cases treated with a CHOP-like protocol [100,123,126]. Two papers with 2 groups and 102 cases used a MOPP-type protocol [79,100], and the remainder had various protocols used [91,159,176]. The group sizes ranged from 13 to 127. Seven of the papers with seven groups reported remission, and seven papers with eight groups included survival times. The outcomes are listed in Table 3.

Single agent chemotherapy was reported in 35 papers with 39 groups and 1230 cases of dogs with generalised lymphoma. Of these, 13 papers with 16 groups and 466 cases described the treatment of naïve lymphoma [30,49,63,74,77,103,108,115,121,125,127,131,177], 13 papers with 13 groups and 455 cases described relapsed lymphoma cases [39,44,46,53,115,138,139,141,143,144,145,147,149], 4 groups and 135 cases from 4 papers included a mix of naïve and relapsed lymphoma cases [152,155,177,178], with the remainder not clearly indicating previous treatment as an exclusion criteria [35,59,61,157,162,163]. Doxorubicin was the most used agent and was used in 19 groups from 16 papers [30,35,49,61,63,74,77,103,108,122,125,139,157,162,163,178]. Mitoxantrone was used in four groups from three papers [115,144,177], CCNU in three groups from three papers [39,46,127], actinomycin-D in two groups in two papers [53,138], cyclophosphamide in two groups from two papers [59,131], etoposide in two groups in two papers [152,155], and single groups of bleomycin [149], carboplatin [141], DTIC [44], L-asparaginase [121], melphalan [145], rabacfosadine [147], and vinblastine [143]. Given the range of drugs used and the combination of use in naïve and relapsed lymphoma cases, summary data are only presented for doxorubicin.

The complete remission rate for doxorubicin-treated dogs was reported for 14 groups and ranged from 22 to 82%, with a median of 65.5% and mean of 61%. Partial remission rates were reported in eight groups and ranged from 0 to 56%, with a median of 12.5% and mean of 17%. The overall response rate ranged from 33 to 94% with a median of 79.5% and mean of 73%. The median remission time was reported for 15 groups, ranging from 14 to 246 days, with a median of 147 and mean of 140 days. The reported mean remission time in four groups ranged from 156 to 203 days, with a median of 180 and mean of 180 days. Median survival times were reported for 13 groups, ranging from 100 to 423 days, with a median of 230 and mean of 237 days. Mean survival times were reported for five groups and ranged from 198 to 351 days. One group from one paper had relapsed lymphoma cases [139]; the remainder were either naïve or prior treatment was not clearly indicated. 

There were 25 papers with 27 groups and 1116 cases of relapsed generalised lymphoma where outcomes for chemotherapy treatment were reported [33,37,39,44,45,46,47,48,50,53,115,137,138,139,140,141,142,143,144,145,146,147,148,149,150]. These were treated with a variation in treatment options including combination and single agent treatments. One study was limited to B-cell lymphoma [147], one included only second relapse cases [143], and the remainder were a mix of immunophenotypes and treatments. The complete remission rate was indicated for 25 groups and ranged from 0 to 65% with a median of 27% and mean of 27%. The partial remission rate was reported for 23 groups, ranging from 0 to 39% with a median of 20% and mean of 19%. For the 23 groups where an overall response rate was available, this ranged from 0 to 77% with a median of 43% and a mean of 44%. 

### 3.4. Remission Times, Survival Times, and Response Rates for Other Forms of Lymphoma Treated with Chemotherapy

Eleven papers reported on lymphoma that was described as neither multicentric nor nodal and treated with chemotherapy, including six reporting outcomes for cutaneous lymphoma [34,42,65,66,67,68,179], predominately cutaneous epitheliotropic T-cell lymphoma (CETL), four with gastrointestinal lymphoma [41,54,69,70], one with hepatic [89], and one nasal [18]. 

In the six papers reporting on cutaneous lymphoma, two used CCNU, one masitinib, one an experimental drug VDC-1101, one the combination VELCAP-EL based on a CHOP-like protocol with additional alkylating agents or CCNU, and one had a mix of treatment protocols. Remission rates were reported in all papers. The CR rate ranged from 0 to 64% with a median of 17% and mean of 22%. The PR rate ranged from 27 to 100% with a median of 50% and mean of 51%. The ORR ranged from 44 to 100% with a median of 78% and mean of 73%. The MRTs reported in five papers ranged from 38 to 106 days, with a median of 85 and mean of 72 days. Survival times ranged from 130 to 281 in the two papers where this was reported, with a median of 207 and mean of 205 days.

In the four studies reporting on gastrointestinal lymphoma, one was limited to colorectal disease, one was intestinal and the final two had predominately gastrointestinal involvement. In one the treatment was VELCAP-SC, one used single agent L-asparaginase, and the other two involved several different protocols. Remission rates were reported in two studies with one study having a CR rate of 50% and a PR rate of 6% and the other a CR rate of 47% and a PR rate of 9%. The MRT ranged from 50 to 1318 days, and the MST ranged from 77 to 1845 days. The studies with colorectal lymphoma had the longest remission and survival outcomes (1318 days and 1845 days, respectively); removing this paper, the ranges were 50 to 86 days and 77 to 147 days for remission and survival, respectively. 

### 3.5. Changes in Remission and Survival Durations over Time

As the papers available for review covered nearly five decades, from 1973 to 2021, the authors looked at the change over time in reported remission and survival durations. Generalised lymphoma was selected as this represents the most frequent presentation of lymphoma. A similar evaluation for B-cell lymphoma was undertaken as the number of groups of T-cell lymphoma were small. The results are shown graphically in Figure 1, Figure 2, Figure 3 and Figure 4.

A median first remission duration was available from 99 papers and a median overall survival from 109 papers for generalised lymphoma. When restricted to B-cell lymphoma, a median first remission duration was available for 19 papers and a median overall survival for 18 papers. The trend lines show a slight increase in the median first remission duration over time (r = 0.183, *p* = 0.070), and a steady median overall survival time (r = 0.002, *p* = 0.982) when all generalised lymphoma types are included. When the results are restricted to papers reporting B-cell lymphoma outcomes, both the median first remission duration (r = −0.053, *p* = 0.829) and median overall survival (r = −0.161, *p* = 0.523) showed a slight decreasing trend over time. For both data sets, there was significant correlation between remission and survival (0.809, *p* < 0.001 for generalised lymphoma; 0.788, *p* < 0.001 for B-cell). 

### 3.6. Prognostic Factors

Prognostic factors for either remission times, survival times, or both were evaluated in 112 papers involving 8933 cases. Each paper reported on the effects of between one and 13 factors. There were 73 papers that evaluated the factors for both remission and survival, 16 evaluated remission duration only, and 22 evaluated survival duration only. One paper reported factors associated with survival for longer than two years. Statistical analysis was univariate in 52 papers and multivariate in 60 papers. The number of cases in the papers ranged from 11 to 775 with a mean of 80 +/− 8.7 and a median of 58.

There were 60 papers in which multivariate analysis results were available, which included 5975 cases. Four of these papers reported on specific locations of lymphoma [28,65,89,180], or in one paper, advanced indolent lymphoma [181]; as these might have differing prognostic factors they were excluded from further discussion. 

The 55 papers that reported prognostic factors based on multivariate analysis for generalised lymphoma were published between 1981 and 2021, including 5706 cases. The number of cases included in each ranged from 18 to 775, with a median of 73 and a mean of 103.75 +/− 16.32. The number of factors included in the analysis ranged from 3 to 17, with a median of 8 and a mean of 8.49 +/− 0.36. For each of the factors included in at least one paper, the number of papers in which they were included ranged from one to 47, with a median of six and a mean of 12.29 +/− 2.28. Remission and survival were evaluated in 40 papers, remission only in seven, survival only in seven, and survival at 2 years in one paper [182]. Only cases with B-cell lymphoma were included in five papers [55,72,102,183,184], and only T-cell lymphoma in four papers [79,91,100,116]. 

Information for variables that were included in fewer than ten papers is included in the Appendix A. There were 14 papers [72,77,91,97,98,124,185,186,187,188,189,190,191,192] that included molecular, genetic, cytochemical, or similar factors. As variables were evaluated in at most two papers, the results are provided in the Appendix A. Treatment was assessed as a prognostic variable in 18 papers. The changes, including protocol variations and drug substitutions, were varied so could not be analysed as a group. A table of treatment parameters analysed is included in the Appendix A. The findings from papers reporting only univariate analysis are included in the Appendix A. 

It was noted that there is inconsistency in the criteria used to include variables in the multivariable analysis, or this was missing in some papers’ material and methods. When described, some papers included variables that were significant at *p* < 0.05 on univariate analysis, whereas others included variables that had a *p*-value of <0.25 on univariate analysis. 

#### 3.6.1. Patient Characteristics

Age was evaluated as a prognostic variable in 40 papers involving 3615 cases. Age was included as a continuous variable in some papers or as age ranges. Both remission and survival were evaluated in 28 papers, remission only in five, survival only in six, and one had survival beyond two years as the outcome measure. Age was identified as being prognostic for both remission and survival times in one paper, remission in three papers, survival in six papers, and not prognostic in 30 papers. Including only papers where it was evaluated, age was prognostic for remission in 12.1% of papers and for survival in 20% of papers.

Patient weight was included in 38 papers as a prognostic variable involving 3978 cases. Both remission and survival were evaluated in 27 papers, remission in five, survival in five, and survival at 2 years in one paper. Weight was prognostic in five papers for both outcomes, remission in two, and survival in two, and was not prognostic in 30 including one paper that looked at body condition score rather than body weight. In the papers that included weight as a variable, it was prognostic for remission in 18.8% and for survival in 24.2%. 

The patient’s sex was evaluated as a variable in 43 papers with 4283 cases. Remission and survival were assessed in 32 of these, remission in five, survival in five, and survival at 2 years in one paper. Sex was identified as prognostic for both remission and survival in two papers, prognostic for survival in four papers, and not significantly associated with outcome measures in the remaining 37 papers. This equates to sex being prognostic for remission in 5.4% and survival in 15.8% of papers in which it was included. 

Neuter status was assessed as a prognostic variable in 10 papers with 681 cases. There were eight papers that looked at both remission and survival with the remaining two only looking at survival. None of the papers identified neutering as being prognostic for the evaluated outcomes. 

Breed was evaluated in 17 papers including 1194 cases. There was a mixture of evaluations of specific breeds such as the boxer, while others compared purebred to crossbred dogs. Both remission and survival were evaluated in 13 papers, remission only in two, and survival in two papers. Breed was prognostic for remission in one paper and survival in one paper, which equates to 6.7% of papers where it was included as a variable.

Substage was included as a variable in 44 papers involving 4524 cases. Evaluation of the effects on both remission and survival was included in 32 papers, remission alone in seven papers, and survival alone in five papers. Substage was found to be prognostic for both outcomes in eight papers, for remission in seven, and survival in five papers. It was found to be prognostic for remission in 38.5% of papers and for survival in 35.1% of papers where this was evaluated. 

Exposure to prednisone/prednisolone prior to chemotherapy was evaluated in 17 papers that included 2538 cases. It was evaluated as affecting both outcomes in 12 papers, remission in four, and survival in one paper. Prior prednisone/prednisolone exposure was prognostic for remission in two papers and for survival in two papers; 12.5% and 15.4%, respectively, of the papers where it was included. 

#### 3.6.2. Lymphoma Characteristics

Only clinical stage, anatomical location, and immunophenotype were included in more than ten papers. The other relevant parameters: grade, histological classification, cell size, and mitotic count are included in Appendix A. 

Clinical stage was an included variable in 47 papers involving 5030 cases. There was inconsistency in the evaluation between papers, some comparing stage V to other stages, some looking at lower stages (I +/− II) compared to others, and some papers including selected stages. Evaluation was undertaken for both remission and survival in 34 papers, remission only in seven, and survival only in six. A significant influence on both outcomes was found in three papers, for remission in three papers, and survival in eight papers. Clinical stage was prognostic for remission in 14.6% of papers and for survival in 27.5% of papers where included. 

The anatomical location was evaluated as a prognostic variable in 13 papers involving 1139 cases. The most common anatomical locations where involvement was reported were the anterior mediastinum or bone marrow. Eight papers included both remission and survival, remission alone was evaluated in three, and survival in two. It was found to be prognostic for both outcomes in three papers and for survival in one. This represents 27.3% of papers when included for remission and 40% for survival. 

The prognostic effects of the immunophenotype of the lymphoma was included in 28 papers with 3926 cases. Both remission and survival outcomes were evaluated in 21 papers, remission alone in four, and survival in four. Immunophenotype was prognostic for both outcomes in seven papers, remission in six, and survival in three papers. It was prognostic for remission in 52% and survival in 41.7% of papers that included these outcomes.

#### 3.6.3. Laboratory Findings

Anaemia, hypercalcaemia, and thrombocytopenia were the only parameters evaluated in more than 10 papers. The remain parameters are included in the Appendix A. 

Anaemia was included as a variable in 22 papers involving 1874 cases. Anaemia was most often reported as categorical data, present or not. It was evaluated as a prognostic variable for both remission and survival in 14 papers, for only remission in three, and survival in five. Anaemia was prognostic for remission in two papers and for survival in six, being 11.8% and 31.6%, respectively, of papers where evaluated. 

The presence of hypercalcaemia was evaluated in 28 papers with 2838 cases. The effects on both remission and survival were included in 22 papers, remission only in four, and survival in two. It was found to be significantly prognostic for both outcomes in three papers, for remission in one, and for survival in two papers. This equates to being prognostic for remission in 11.8% and survival in 31.6% of papers where included. 

Thrombocytopenia was included as a prognostic variable in 13 papers with 1323 cases. Both remission and survival were included in nine papers, remission alone in three, and survival in one. It was found to be prognostic for remission in two and survival in three papers. Thrombocytopenia was identified as a positive prognostic factor for survival in one paper [193]. When included, it was prognostic in 16.7% of papers for remission and 30% of papers for survival. 

#### 3.6.4. Treatment Parameters

The response to treatment, most often indicated as the patient achieving complete remission, was included in 10 papers involving 1211 cases. Both remission and survival were included in eight papers, remission alone in one, and survival alone in one paper. It was identified as prognostic for both outcomes in five papers and for survival in three. It was prognostic in 62.5% of papers for remission and 88.9% for survival when included. 

## 4. Discussion

This review shows there is a large degree of variability in the literature for outcomes for lymphoma and prognostic variables. Variability in the results of the papers have arisen from the variable inclusion/exclusion criteria, the variety of definitions used, the analyses performed, and potentially the number of cases included. Despite there being many papers in the literature, it is difficult to acquire a clear view of the topic or evidence to inform best practice. 

A basic measure in dogs with lymphoma treated with chemotherapy is response. We described the literature in which the criteria used to assess this fundamental feature is inconsistent, including when it is assessed, and the duration required to be considered a response. Prior to 2010, a veterinary scheme for describing responses in nodal lymphoma was lacking, which was addressed in the VCOG review. However, since the VCOG paper [7] was published, the proposed uniform approach has not been consistently utilised. The response rates reported vary greatly, with CR rates for nodal lymphoma treated with a CHOP-like protocol ranging from 25 to 100%. Wide variability persisted even when we selected only papers that use comparable response classification schemes including criteria for response and duration for which it needed to be sustained (data not shown), suggesting factors other than the response criteria definition had a role in the variation.

A key feature in the decision making in lymphoma is first remission duration rather than overall survival. Survival is affected by other factors such as the owners’ willingness to pursue repeated chemotherapy, for financial or logistical reasons, and the availability of euthanasia for veterinary patients as an option. There is varied terminology and methodology used to report remission duration. The initial date used ranged from the date of diagnosis through to completion of a fixed portion of a protocol. This could be a difference of 60 days when studies have a mean and median reported MRT of about 200 days. This would contribute to some of the variability in outcomes, but with a wide range of median remission times, from 73 to 622 days for nodal lymphoma treated with CHOP-like chemotherapy, there are likely other factors that contribute. The VCOG paper [7] has a recommendation that these intervals be taken from the initiation of treatment, which would remove this variability. The consensus had a recommendation for the timing of rechecks during and after therapy to identify the progression of disease [7], which was often not documented in past studies with many being retrospective. 

The variability in the survival times reported is due to factors that approximate those mentioned regarding remission such as the definition of the duration’s starting point. While survival times were longer than remission times by approximately 50%, a different starting point could account for about 10% of the result. As mentioned above, survival was affected by the owner’s decisions on further treatment or euthanasia. 

When clearly defined specific subtypes or anatomical locations are considered, which would better match the information from people, the number of papers and included cases decreases. The most common form of lymphoma in dogs is nodal or multicentric DLBCL. There are only 11 papers that included groups confined to B-cell lymphoma, of which only five were restricted to DLBCL. As shown in Table 3, a wide range of outcomes have been reported for both B-cell and T-cell lymphoma that potentially arise from different subtypes being included in the studies. 

The change in outcome over time as measured by either remission duration or survival has not changed in a clinically significant degree as indicated in Figure 1, Figure 2, Figure 3 and Figure 4. In fact, there is a trend of declining outcomes for B-cell lymphoma, which could be due to several reasons such as owners and veterinarians being more willing to treat patients who are more unwell or with increased numbers of negative prognostic factors, as the treatments used have remained relatively constant. 

There is a reasonable body of literature evaluating variables as prognostic indicators for outcome for lymphoma in dogs. However, we found variability in the study design and how the factors were evaluated. There were more than 45 factors that were included in at least one paper, as we grouped some related ones together, but the most evaluated in any single paper was 17, with some including as few as three. This is confounded by the differing criteria used as such for the inclusion of variables in the multivariate analysis. The limited inclusion of prognostic factors in many of the studies has the potential to bias the results of the analysis if a confounding variable is not included. We did not detail the results of univariate analysis in the manuscript as these did not remain significant when multivariate analysis was performed with a reasonable frequency, and lack of the latter makes interpretation difficult. 

What we found interesting was that the majority of the commonly cited prognostic factors were prognostic in less than 50% of the papers in which they were evaluated, many in less than 20%. If we exclude treatment type and molecular/genetic markers as diverse groups, and factors evaluated in fewer than 10 papers, only immunophenotype and response to treatment were prognostic in greater than 50% of the papers evaluated, and immunophenotype only for remission, and 41.7% for survival. 

Hypercalcaemia and substage are commonly cited as strong predictors of outcome for dogs with lymphoma, but these were identified as prognostic in 38.5% and 11.8% of papers, respectively, for remission and 35.1% and 31.6% of papers, respectively, for survival. We must question whether this has arisen from reliance on a small number of studies, or a weakness in the methods of the studies that did not show these to be prognostic indicators. There is variability in the assessment by veterinarians on substage [194], which cannot be accounted for in this review. Anatomical location, predominately relating to the presence of anterior mediastinal disease, was another of the more frequently identified prognostic factors. Many of the earlier studies did not include immunophenotype, which has an association with anterior mediastinal disease in dogs, and if both are included, the immunophenotype might be the prognostic indicator. If gastrointestinal disease is assessed, this has a reported poorer outlook than multicentric or nodal disease [41], highlighting the need to consider anatomical presentation in addition to other classifications. 

The way prognostic indicators were evaluated was not consistent. The clinical stage of lymphoma was prognostic for remission in 14.6% of papers and for survival in 27.5% of papers. Some papers included stage I disease, and as would be expected, these dogs did better, so overall stage was reported as prognostic. Other excluded high-grade disease, stage V, and did not find stage to be prognostic, possibly missing the fact that stage V might have a worse outlook. There is no information on the influence of stage V lymphoma on outcomes when differences in how this was diagnosed are considered. Bone marrow involvement might differ from the involvement of a non-lymphoid organ, or the degree of bone marrow involvement might influence outcome.

Disappointingly, there were potentially prognostic variables that were evaluated in few studies. Some, such as dogs experiencing toxicity or dose modification, are often cited as being prognostic, but these were included in only five and six papers, respectively. The histological classification was only evaluated in eight papers but was prognostic for survival in 42.9% of papers in which it was included. Histology is being performed less often with the common use of cytology, immunocytochemistry, and flow cytometry to diagnose and classify patients. The failure to include a reasonable range of factors when assessing prognosis might miss important information that could influence treatment choices and decisions when they are omitted, or potentially bias statistical analysis.

## 5. Conclusions

The authors propose the use of the VCOG [7] recommendation for the classification of response for nodal lymphoma to be used in all future studies, including the recommendations for the definition of remission and survival. Additionally, a standardised protocol for follow-up is recommended [7] to be used both in practice, and in all prospective studies. Consistency of information in patient records could allow metanalysis to be performed across studies in order to gain a better understanding of the factors that influence the outcome for patients with lymphoma, as has been recommended in the review by the European Canine Lymphoma Network [6]. 

Further work is required to develop a consensus on the prognostic indicators that should be included in lymphoma studies and if the current staging system needs refinement for use in outcome studies. Having a consensus on substage classification for dogs with lymphoma is required as misclassification included in the statistical analysis has an unknown effect. When evaluating prognostic factors, future studies should endeavour to include a reasonable number of factors in the analysis and not exclude potentially confounding factors. 

## Figures and Tables

**Figure 1 vetsci-10-00342-f001:**
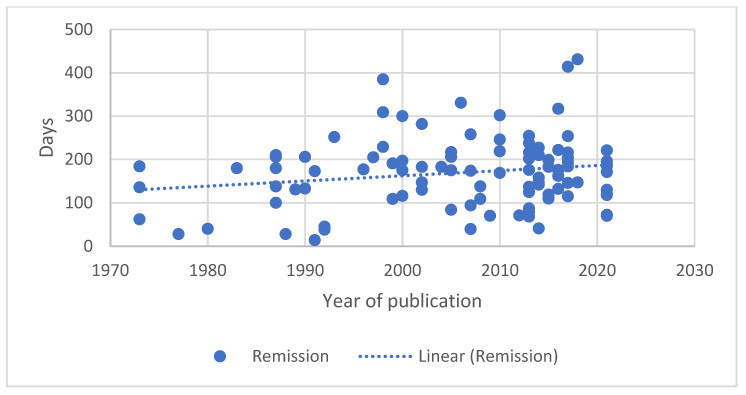
Median remission duration for dogs with generalised lymphoma treated with chemotherapy plotted against date of paper publication.

**Figure 2 vetsci-10-00342-f002:**
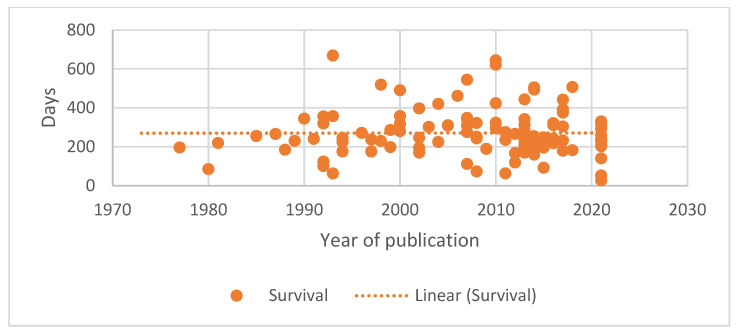
Median survival duration for dogs with generalised lymphoma treated with chemotherapy plotted against date of paper publication.

**Figure 3 vetsci-10-00342-f003:**
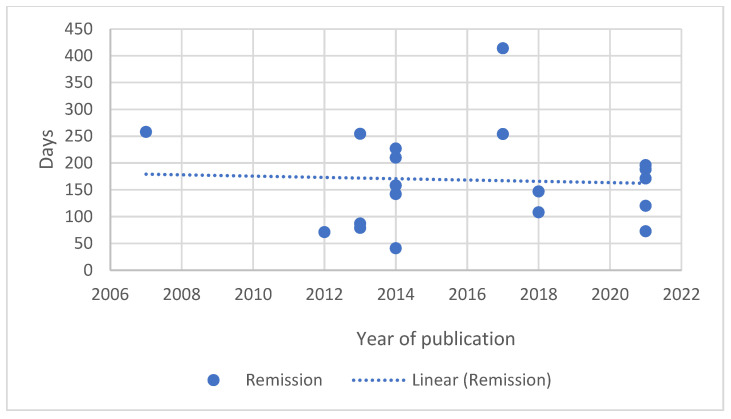
Median remission duration of dogs with generalised B-cell lymphoma treated with chemotherapy plotted against date of paper publication.

**Figure 4 vetsci-10-00342-f004:**
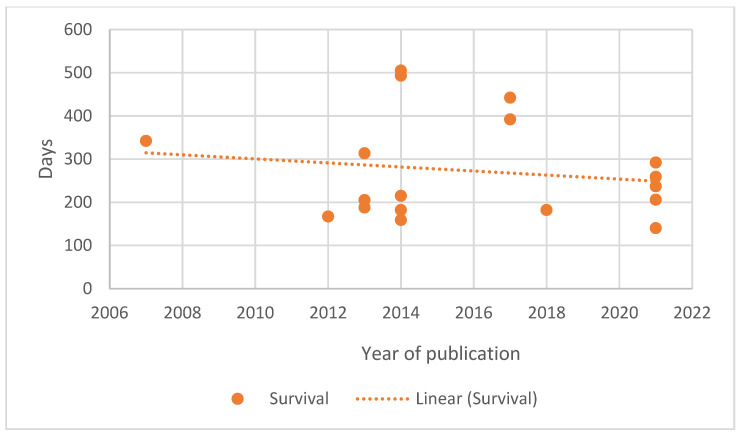
Median survival duration of dogs with generalised B-cell lymphoma treated with chemotherapy plotted against date of paper publication.

**Table 1 vetsci-10-00342-t001:** Response criteria definitions used in papers.

Treatment Response Criteria Reported	Number of Papers	CR Definition (n)	PR Definition (n)	PD Definition (n)
CR only	3	100% decrease (2) Not defined (1)	N/A	N/A
CR, PR, NR	38	100% decrease (33) 75% decrease (1) 50% decrease (1) Not defined (3)	50% decrease (39) 25–50% decrease (1) Not defined (8)	Not CR or PR (38)
CR, PR, SD, PD	50	100% decrease (50)	50% decrease (39) 30% decrease (3) Not defined (8)	50% increase (8) 25% increase (19) 20% increase (4) 10% increase (3) Not defined (16)
VCOG or modified RECIST	30	100% decrease (30)	30% decrease (30)	20% increase (30)
Not stated	35	N/A	N/A	N/A

CR = complete remission, PR = partial remission, SD = stable disease, PD = progressive disease, NR = no response, VCOG = veterinary comparative oncology group, RECIST = response evaluation criteria in solid tumours.

**Table 2 vetsci-10-00342-t002:** Remission and survival terms used in the papers. The preferred terms from VCOG [5] are progression-free survival and lymphoma-specific survival.

Outcome Term	Number of Papers	Variations Included in the Term in This Review
**Remission times**		
Progression-free survival	50	Disease-free interval Progression-free interval
Duration of remission	30	Duration of first remission Time of first remission Relapse-free interval Remission time
Median response period	32	Median duration response Median response time Median response duration
Objective response duration	1	Objective response time
Time to progression	18	Time to failure Time to relapse
Not available	25	
**Survival times**		
Overall survival time	101	Median survival time
Lymphoma-specific survival	11	
Length of survival	1	
Not available	43	

**Table 3 vetsci-10-00342-t003:** Outcomes reported for dogs with multicentric or nodal lymphoma treated with CHOP-like chemotherapy.

Outcome	CHOP-like, Multicentric or Nodal Lymphoma (Number of Groups)	CHOP-like, Anatomical Location Not Clearly Defined Lymphoma (n = Number of Groups)	COP-like, Mixed Lymphoma (n = Number of Groups)	CHOP-like, B-Cell Multicentric or Nodal lymphoma (n = Number of Groups)	T-cell Multicentric or Nodal Lymphoma (n = Number of Groups)
Number of papers	43 (54)	26 (33)	21 (27)	11 (14)	8 (9)
CR %	25–100 (45)	27–100 (24)	53–90 (23)	63–100 (5)	64–92 (4)
Mean CR %	76	76	71	79	82
Median CR%	77	78	73	77	86
PR %	0–33 (34)	4–21 (19)	2–40 (21)	0–15 (3)	8–11 (3)
Mean PR %	13	12	18	6	9
Median PR%	9	12	15	4.5	9
ORR	64–100 (34)	27–100 (19)	56–100 (21)	64–100 (3)	73–96 (3)
Mean ORR	86	85	88	83	88
Median ORR	91	89	90	87	94
MRT range in days	41–414 (35)	28–331 (23)	28–199 (20)	41–414 (14)	118–431 (5)
MRT mean in days	199	195	125	187	237
MRT median	196	208	128	192	175
Mean remission in days	59, 518 (2)	77, 201 (2)	31–323 (6)	59 (1)	NR
MRT CR in days	104–317 (13)	202–348 (10)	132–215 (3)	NR	104–316 (3)
MRT PR in days	43–117 (3)	41 (1)	75 (1)	NR	60 (1)
MST range in days	73–622 (44)	175–644 (23)	85–357 (15)	159–505 (14)	26–507 (8)
MST mean in days	287	333	226	297	200
MST median	267	314	224	276	191
Mean survival in days	136 (1)	207–459 (3)	138–386 (8)	136 (1)	NR
MST CR in days	244–573 (4)	309–446 (3)	266, 290 (2)	NR	362 (1)
MST PR in days	50–177 (7)	97–205 (3)	129, 152 (2)	NR	177 (1)
% Remission at 1 year	17–46 (6)	26 (1)	0–19 (3)	NR	27, 54 (2)
% Remission at 2 years	12–35 (6)	NR	0, 5 (2)	NR	16, 29 (2)
% Remission at 3 years	7–13 (3)	NR	0 (2)	NR	12, 13 (2)
% Survival at 1 year	28–70 (10)	22–57 (7)	0–40 (6)	NR	31, 54 (2)
% Survival at 2 years	6–33 (8)	11–26 (4)	0–17 (6)	NR	20, 35 (2)
% Survival at 3 years	12–25 (3)	4 (1)	0–15 (5)	NR	12, 12 (2)

NR = not reported for any group. CR = complete remission, PR = partial remission, ORR = overall response rate, MRT = median remission time, MST = median survival time.

## Data Availability

There was no new data generated in this review that has not been presented in the manuscript or Appendix A.

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
