# Peer review of "Review of Canine Lymphoma Treated with Chemotherapy—Outcomes and Prognostic Factors"

_vetsci, 2023, doi:10.3390/vetsci10050342_

Round 1

Reviewer 1 Report

Thank you for taking this critical review for lymphoma study. A more updated consensus should warrant better lymphoma management.

Author Response

Dear Reviewer,

Thank you for these comments that support our suggestion of an updated consensus on reporting this common disease.

Regards

Peter, Peter and Rosanne

Reviewer 2 Report

Dear Authors,

thank you for your work, which gives a clear overview. Works like this should be a starting point to highlight critical issues and improve reporting and possibly also patient management.

A similar study was conducted a few years ago by the European Lymphoma Network Group and should be quoted (Marconato L, Polton GA, Sabattini S, Dacasto M, Garden OA, Grant I, Hendrickx T, Henriques J, Lubas G, Morello E, Stefanello D, Comazzi S; European Canine Lymphoma Network. Conformity and controversies in the diagnosis, staging and follow-up evaluation of canine nodal lymphoma: a systematic review of the last 15 years of published literature. Vet Comp Oncol. 2017 Sep;15(3):1029-1040. doi: 10.1111/vco.12244. Epub 2016 Jul 14. PMID: 27412493.)

Also, the Valli paper (Valli VE, Kass PH, San Myint M, Scott F. Canine lymphomas: association of classification type, disease stage, tumor subtype, mitotic rate, and treatment with survival. Vet Pathol. 2013 Sep;50(5):738-48. doi: 10.1177/0300985813478210. Epub 2013 Feb 26. PMID: 23444036.) should be listed in the Introduction next to reference 3.   Thank you again, very nice job!  

Please edit Simple Summary and Abstract.

Some minor editing is needed also in the Introduction.

Author Response

Dear Reviewer,

Thank you for your review and comments.  

We have edited the simple summary, abstract and introduction as you have suggested.  These have been made using track changes in the attached extract and the revised manuscript.  

Thank you for highlighting our oversight of the references by Valli and Marconato.  The Valli reference has been added in the introduction next to reference 3 as suggested.  The Marconato reference has been included in the introduction as an example of a critical review and included in the conclusion as support for the need to have better consistency in the collected data and reporting in studies on lymphoma.  

Kind Regards

Peter, Peter and Rosanne.

Reviewer 3 Report

Interesting and comprehensive review of a particularly complicated topic.

Authors made a significant review of the literature and additional, substantial changes are not required.

Author Response

Dear Reviewer,

Thank you for these comments and your review.

Regards

Peter, Peter and Rosanne.